# Abstracts of Presentations to the Working Session on Improving Predictive Modeling of Mycotoxin Risk for Africa Held at the 3rd ASM2022 on 7 September 2022, in Stellenbosch, South Africa

**DOI:** 10.3390/toxins15030174

**Published:** 2023-02-24

**Authors:** Felix Rembold, Brighton Mvumi, David Miller, Rose Omari, Paola Battilani, Yamdeu Joseph Hubert Galani, Wiana Louw, Titilayo D. O. Falade, Wolfgang Schweiger, Monica Ermolli

**Affiliations:** 1European Commission Joint Research Centre (EC JRC), Via E. Fermi 2749, 21027 Ispra, VA, Italy; 2Department of Soil Science and Agricultural Engineering, Faculty of Agriculture, University of Zimbabwe, 630 Churchill Ave., Harare P.O. Box MP167, Zimbabwe; 3Department of Chemistry, Carleton University, 1125 Colonel By Drive, Ottawa, ON K1S 5B6, Canada; 4Science and Technology Policy Research Institute, Council for Scientific and Industrial Research (CSIR-STEPRI), Accra P.O. Box CT 519, Ghana; 5Department of Sustainable Crop Production, Università Cattolica del Sacro Cuore, Via Emilia Parmense, 84, 29122 Piacenza, PC, Italy; 6Section of Natural and Applied Sciences, Canterbury Christ Church University, N Holmes Rd., Canterbury CT1 1QU, UK; 7Southern Africa Grain Laboratory (SAGL), 477 Witherite St., The Willows, Pretoria 0040, South Africa; 8International Institute of Tropical Agriculture, West Africa Hub, PMB 5320, Oyo Road, Ibadan 200001, Oyo State, Nigeria; 9BIOMIN Research Center, Technopark 1, 3430 Tulln, Austria

**Keywords:** mycotoxins, aflatoxin, fumonisin, modeling, data input, mapping, warning, cereal, Africa

## Abstract

In 2008, the African Postharvest Losses Information Systems project (APHLIS, accessed on 6 September 2022) developed an algorithm for estimating the scale of cereal postharvest losses (PHLs). The relevant scientific literature and contextual information was used to build profiles of the PHLs occurring along the value chains of nine cereal crops by country and province for 37 sub-Saharan African countries. The APHLIS provides estimates of PHL figures where direct measurements are not available. A pilot project was subsequently initiated to explore the possibility of supplementing these loss estimates with information on the aflatoxin risk. Using satellite data on drought and rainfall, a time series of agro-climatic aflatoxin risk warning maps for maize was developed covering the countries and provinces of sub-Saharan Africa. The agro-climatic risk warning maps for specific countries were shared with mycotoxin experts from those countries for review and comparison with their aflatoxin incidence datasets. The present Work Session was a unique opportunity for African food safety mycotoxins experts, as well as other international experts, to meet and deepen the discussion about prospects for using their experience and their data to validate and improve agro-climatic risk modeling approaches.

## 1. Preface

Since 2008, the African Postharvest Losses Information Systems project (APHLIS, www.aphlis.net, accessed on 6 September 2022) has been reporting postharvest cereal loss estimates across 37 sub-Saharan African countries. A pilot initiative was undertaken to supplement the loss estimations with information on mycotoxin risks. In particular, the possibility of monitoring agro-climatic conditions favorable to *Aspergillus flavus* growth in near-real time for maize has been explored to identify periods and areas of increased risk of cereal contamination based on the existing knowledge. This led to the production of experimental agro-climatic aflatoxin risk maps with the aim of providing early warning information about the possible increased risk of aflatoxin contamination to guide targeted ground-level contamination surveying and risk mitigation. As evidenced by a recent independent review carried out by a panel of internationally recognized experts, the information produced at this stage is not sufficiently accurate to provide direct aflatoxin agro-climatic risk warnings to countries. To achieve this, crop-specific and validated models would need to be codeveloped and tested collaboratively with practitioners in Africa, and the feasibility of such an approach is still difficult to evaluate and needs more testing. Preliminary online meetings were organized in April 2022 with experts who have carried out significant mycotoxin-related research in five African countries, including Ghana, Senegal, South Africa, Tanzania, and Benin, and initial datasets and collaboration opportunities have been identified.

The present Work Session was a unique opportunity for those African food safety mycotoxin experts and other international experts to meet and deepen the discussion about the prospects for using their experience and their data to:
Determine the relevance and usefulness of the agro-climatic aflatoxin risk maps for selected periods and geographic areas in Africa.Propose approaches to compare existing mycotoxin incidence data and biomarkers with agro-meteorological data and, where possible, undertake such studies.

## 2. Introduction to the APHLIS Working Session

### 2.1. Introduction to the APHLIS Project

MvumiBrighton[Aff af11-toxins-15-00174]Department of Soil Science and Agricultural Engineering, Faculty of Agriculture, University of Zimbabwe, 630 Churchill Ave, Harare P.O. Box MP167, Zimbabwe*Correspondence: mvumibm@hotmail.com

The APHLIS project (www.aphlis.net) has been studying postharvest cereal losses estimations in Africa for almost 14 years and has become the main scientific reference for estimating weight loss in cereals. Specifically, the project aims at estimating what percentage of the loss occurs at different stages of cereal postharvest management up to marketing. The factors influencing food loss and contamination include conditions and practices during harvesting; drying; transport; threshing; pest population density; and the quality, hygiene, management, and monitoring of the storage facilities alongside the storage duration and conditions. APHLIS considers nine different cereals (maize, sorghum, millet, wheat, barley, rice, teff, fonio (*Digitaria* species), and oats) in 37 sub-Sahara African countries. This involves the integration of seasonal production, storage, and climate data with the postharvest loss profiles for each crop and climate situation, which are based on scientifically measured data. The work of the APHLIS network has made it possible to develop estimates of cereal grain weight losses and the financial and nutritional values and impacts of these losses on the countries and populations. In the last 7 years, the project has been working to include new crops that are key in African diets: legumes (dried cowpeas, beans, and groundnuts) and root and tuber crops (cassava and sweet potatoes). The APHLIS initiative was initially funded by the European Commission, and in 2016, the Bill & Melinda Gates Foundation continued and extended the program through a grant that will finish in June 2023. Initial funding by the European Commission was mainly driven by food security concerns and the low accuracy of postharvest losses in national food balance sheets. With the growing evidence made available by APHLIS, policy and decision-making support has grown. APHLIS is now able, for example, to directly support a number of African countries in their biennial reporting towards the Malabo declaration commitments of halving postharvest losses by 2025. The current phase of the APHLIS project also includes a work package on aflatoxins’ early warning signs, and this is probably the most interesting aspect for this workshop. 

### 2.2. Introduction to APHLIS Agro-Climatic Mycotoxin Risk Warning Model. Verification and Possible Improvement of the Existing APHLIS Agro-Climatic Mycotoxin Risk Warning Model

RemboldFelix[Aff af13-toxins-15-00174]European Commission Joint Research Centre (EC JRC), Via E. Fermi 2749, I-21027 Ispra, VA, Italy*Correspondence: felix.rembold@ec.europa.eu

It is known that several factors are relevant for predicting the growth of *Aspergillus flavus* and *A. parasiticus* on maize and groundnuts: primarily, the weather and insect pests both in the field and during storage. However, quantitative relationships between preharvest and harvest time weather anomalies and aflatoxin occurrence at later stages are practically impossible to measure. The APHLIS approach consists of simple agro-climatic assumptions calculated at the province (primary administrative) level for African countries based on existing early warning approaches in the literature. For the evaluation of the model results, there remain important limitations. The current in silico models account for up to 90% of the concentrations of aflatoxins in maize, but their performance under field conditions is less than 50%. Secondly, data scarcity about aflatoxin occurrence is the main limitation for model calibration and validation at the field level. The data scarcity limitation means model calibration is not yet possible at the country or larger level. Based on the current literature, the approach focuses on an anomaly of meteorological factors during two specific phases of the maize growth cycle during the year: maize grain filling and harvesting. At present, this cannot be extended to other crops. Based on data availability and the literature about the roles of weather during the pre- and postharvest stages, two drivers for warnings were identified. Drought stress during the grain-filling phase of maize and excessive rainfall around harvest. As near-real-time phenology information is not available, the grain-filling phase coincides with the mean maturation period and the harvest time with the senescence period, according to a mean remote sensing-derived phenology. For drought stress values, we use the Anomaly hot Spots of Agricultural Production drought indicator during ripening. For rainfall values, we use an exceptional indicator of rain during “senescence” (SPI > 1.5 SD anomaly). For both drought and rainfall, anomalies > 25% of the agricultural area were considered. For example, the algorithm flagged that, in 2019, in Zambezi Province in Zambia, there was a drought and a potentially increased risk of high aflatoxin concentrations in the maize during maturation. Other examples are East Africa’s first 2020 very rainy crop season, where a high risk of aflatoxin occurrence around harvest was predicted in several countries. We have the records to show historical maps of the frequency of agro-climatic aflatoxin risk for pre-harvest warnings related to drought and excessive rainfall around a crop harvest period over 15 years or more. Currently, agro-climatic warnings can potentially provide experts with early warning information of a possible high risk of aflatoxin contamination.

This pilot aflatoxin risk warning model is based on an expectation that users know that the alerts do not indicate the actual contamination but only the probability that it may happen. There is a high risk of misclassification because of non-weather factors affecting the presence of fungi and mycotoxin production. For example, the presence of insects both in the field and in stored grain is known to increase the risk of the presence of mycotoxins. However, the agro-climatic risk warning maps can help to prioritize targeted contamination testing and risk mitigation measures in combination with other evidence (other factors, postharvest information, etc.) available to experts. These agro-climatic maps, however, are not suitable for use by non-expert personnel. The risk warning thus produced should be combined with other evidence before being made public and used by decision-makers or publicized by journalists. For both the improved validation of the agro-climatic risk warnings of APHLIS and for identifying case studies for improved modeling, close collaboration with African scientists is crucial. Moreover, from the discussions that led to this workshop, we have already seen that there is considerable expertise, and datasets exist in this community that would greatly support both objectives of the workshop. This workshop was also an opportunity to discuss avenues for improved collaboration on mycotoxins research in Africa, which include the standardization of sampling, measurements, and equipment. Additional work on predictive modeling should be achieved on a collaborative basis and possibly support the standardization of mycotoxins data collection and measurements. The first workshop session provided feedback on these questions. For example, whether participants had the time and resources to apply the method for the countries represented, comparing it with local information on the incidence of aflatoxin. The second session involved collecting information about the available data from those countries represented. This can be used to help assess whether the risk modeling generated reliable predictions.

## 3. Invited Lecture

### 3.1. Challenges in Applying Continent-Wide Mycotoxin Prediction Systems in Africa

MillerDavid[Aff af15-toxins-15-00174]Department of Chemistry, Carleton University, Ottawa, ON K1S 5B6, Canada*Correspondence: david.miller@.carleton.ca

A number of researchers have attempted to develop prediction systems for aflatoxin for maize and groundnuts in the field and in storage. None of these has proven to be useful in field applications on a wide scale. Algorithms developed using in vitro data have not translated to real-world conditions. For maize, the planting dates vary according to the microclimate of the farmer’s fields, which challenges assumptions based on area-wide data. Drought is an important factor for aflatoxin; however, insect pressure is also critical. While the former can be estimated with satellite data, data on insect pressure requires the collection of local information. Conditions that favor aflatoxin production in maize in subtropical conditions are different from in dryland conditions. The reliability and predictive power of the results of successful models depend entirely on the quality and number of data points from farmers’ fields coupled with a decade of field experience to refine the model. Similar challenges have been demonstrated in the development of models for groundnuts.

In Africa, another key issue is that recent comprehensive biomarker studies have shown that, in most cases, people are generally co-exposed to fumonisin and, sometimes, high levels of deoxynivalenol and zearalenone. Co-exposure to fumonisin and aflatoxin is synergistic for cancer. Thus, it may be misleading to the consumer to focus on only one toxin. Nonetheless, a number of researchers have advocated making the effort to develop models to provide at least some warning of more versus less risky growing conditions. One way forward is to gather both field and biomarker data from regions in Africa where this exists and work with the respective regional weather and satellite measures of the growing conditions to assess the potential for medium-term success.

**Keywords:** mycotoxins; aflatoxin; risk modeling; Africa

### 3.2. Comparison of APHLIS Aflatoxin Risk Maps with Aflatoxin Data Collected at Harvest in Three Agro-Ecological Zones in Ghana

OmariRose[Aff af16-toxins-15-00174][Aff af19-toxins-15-00174]AgbetiamehDaniel[Aff af17-toxins-15-00174]AnyebunoGeorge[Aff af18-toxins-15-00174]1Science and Technology Policy Research Institute, Council for scientific and Industrial Research, Accra P.O. Box CT 519, Ghana2International Institute for Tropical Agriculture, Accra P.O. Box M32, Ghana3Food Research Institute, Council for Scientific and Industrial Research, Accra P.O. Box M20, Ghana*Correspondence: rose.omari@yahoo.com

The African Postharvest Losses Information Systems (APHLIS) has produced pilot agro-climatic aflatoxin risk maps to provide early warning information due to the risk of aflatoxin contamination to guide risk mitigation. The evidence shows that this pilot model currently has a limited ability to provide direct aflatoxin agro-climatic risk warnings. Thus, the objective of this paper was to assess the extent to which aflatoxin data collected in Ghana matches the indications provided by the APHLIS model by examining the aflatoxin data for maize samples at harvest in 2015 and 2016 in three agro-ecological zones. These included the Savanna (in Brong Ahafo and the northern regions), Humid Forest (in Ashanti and part of the Brong Ahafo Regions), and the Southern Guinea Savanna (in the upper east and upper west regions). In these regions, the APHLIS map did not show any warning in the forest and transition zones, though the aflatoxin levels were 15 and 21 μg/kg during the minor and major harvest periods, respectively. The map, however, showed pre-harvest drought stress warnings for maize in 2015 in the three regions of the Savannah zone, with aflatoxin levels of 98 μg/kg at harvest in one region and 6.3 and 4.7 μg/kg in the other two regions. Furthermore, the map showed no warning at pre-harvest in 2016, but the mean aflatoxin levels at harvest ranged from 122 to 301 μg/kg. These findings suggest that the APHLIS model could predict the risk of aflatoxin contamination but only to some extent. Thus, in addition to the climatic conditions, other factors such as on-field good or bad practices that could contribute to aflatoxin production or reduction need to be considered in the model to enhance its predictive accuracy. Since aflatoxin contamination is often highest at the postharvest stages, the APHLIS aflatoxin risk warning model could also explore including data on various maize storage techniques under different climatic conditions.

**Keywords:** maize; aflatoxin; warning; mapping; Ghana

### 3.3. Agro-Climatic Risk Warnings for Africa and the European Experience

BattilaniPaola[Aff af21-toxins-15-00174]LeggieriMarco CamardoDepartment of Sustainable Crop Production, Università Cattolica del Sacro Cuore, via Emilia Parmense 84, 29122 Piacenza, PC, Italy*Correspondence: paola.battilani@unicatt.it

Meteorological conditions are the main driving variables for mycotoxin-producing fungi and the resulting contamination in maize grain, but the cropping system can considerably mitigate the weather impact. The weather conditions are the main factors for fungal behavior, and agronomic practices can reduce mycotoxin contamination. This includes pest control, irrigation, and harvest at the proper humidity of grain as examples. Therefore, risk prediction and mapping are excellent support to highlight where the adoption of best practices as described above is critical. APHLIS used weather anomaly information from ASAP (Anomaly Hotspots of Agricultural Production) to predict possible pre-harvest contamination by aflatoxins based on drought stress and excessive rainfall and produced advice at the provincial level for sub-Saharan Africa. Satellite remote sensing data on rainfall and vegetation are used as the input. This is an excellent initiative, based on an empiric approach; the use of remote sensing as the data source, the wide area covered, the release of automatic alerts, and the free access for stakeholders must surely be mentioned as pros of this warning system. This allows comprehensive and homogeneous data input and well-timed output available for all stakeholders. Validation of the predicted data and feedback from the users would add value to the system. At the European level, two weather-based mechanistic models were developed: "AFLA-maize” and “FER-maize”, predicting aflatoxin B1 and fumonisins, respectively. They are based on the infection cycle of the two toxigenic fungi and their interactions with the host plant. The risk of contamination is predicted daily during the growing season, starting from silk emergence, using hourly data of air temperature, humidity, and rainfall. The output reports the probability of producing maize grain contaminated above the legal limit in force in Europe. A recent step forward was a machine learning approach that significantly improved predictions, including cropping system data. They are in use in Italy, but a European-wide system is not yet planned. The pros of mechanistic models remain their flexibility, as they work properly in different geographic areas, and with climate change projections, they run with small-scale data input, and they are extensively validated. Climate change underlined the cooccurrence of fungi, and the urgent need to account for fungi interactions and the development of a joint model for aflatoxin and fumonisin prediction in maize is ongoing. The extension of the predictions to Europe and the support of satellite data would have a great positive impact on the European experience. The “take-home message” is a confirmation of the relevance of predictive models in managing mycotoxins and the benefit in exchanging experiences, which always helps their improvement.

**Keywords:** aflatoxin; fumonisin; modeling; data input; mapping; warning

### 3.4. Maize Aflatoxin Contamination in Regions of Tanzania and Malawi

GalaniYamdeu Joseph Hubert[Aff af22-toxins-15-00174][Aff af23-toxins-15-00174][Aff af31-toxins-15-00174]HindmarchDuncan[Aff af23-toxins-15-00174]MatumbaLimbikani[Aff af24-toxins-15-00174]MartinHaikael David[Aff af25-toxins-15-00174]MouriceSixbert Kajumula[Aff af26-toxins-15-00174]SalluSusannah[Aff af27-toxins-15-00174]KambwiriAlfred Mexon[Aff af28-toxins-15-00174][Aff af29-toxins-15-00174]KuwaliPamela[Aff af29-toxins-15-00174]SongoleAbel Lawrence[Aff af30-toxins-15-00174]KaziVivian[Aff af30-toxins-15-00174]OrfilaCaroline[Aff af23-toxins-15-00174]GongYun Yun[Aff af23-toxins-15-00174]1Section of Natural and Applied Sciences, School of Psychology and Life Sciences, Canterbury Christ Church University, Canterbury CT1 1QU, UK2School of Food Science and Nutrition, University of Leeds, Woodhouse Lane, Leeds LS2 9JT, UK3Natural Resources College, Lilongwe University of Agriculture and Natural Resources, Bunda College Campus S125 Road, Lilongwe P.O. Box 219, Malawi4Department of Food Biotechnology and Nutritional Sciences, Nelson Mandela African Institution of Science and Technology, Old Moshi Road, Nambala, Arusha 23311, Tanzania5Department of Crop Science and Horticulture, Sokoine University of Agriculture, Morogoro P.O. Box 3000, Tanzania6Sustainability Research Institute, School of Earth and Environment, Faculty of Environment, University of Leeds, Leeds LS2 9JT, UK7Centre for Environmental Policy and Advocacy, Blantyre P.O. Box 1057, Malawi8Civil Society Agriculture Network, Lilongwe P.O. Box 203, Malawi9Economic and Social Research Foundation, Dar es Salaam, Ursino Estate P.O. Box 31226, Tanzania*Correspondence: joseph.galaniyamdeu@canterbury.ac.uk

Aflatoxins are one of the most harmful mycotoxins contaminating maize in sub-Saharan Africa, affecting the health and economy of the populations. For mitigating the risks linked to aflatoxin contamination, it is important to understand its level in foods and its contributing factors. This study aimed to determine the level of aflatoxins in maize samples collected from different regions of Tanzania and Malawi and access the socioeconomic, farming, and storage parameters influencing aflatoxin contamination. Maize grains were collected in households during the harvest season in 2019 (142 samples in Tanzania and 87 in Malawi) and in 2021 (126 and 85 samples). Additionally, 84 samples were collected six months after harvest in Malawi (this sampling could not be performed in Tanzania due to COVID-19 travel restrictions). The grains were ground to flour, and aflatoxins B1, B2, G1, and G2 were analyzed in maize flour by high-performance liquid chromatography (HPLC) coupled with fluorescence detection (FLD).

In Tanzania, aflatoxin B1 showed the highest occurrence (7.8% of the samples above the limit of quantification in 2019 and 28.6% in 2021), and in both years, around half of the contaminated samples had AFB1 and total aflatoxins levels above the regulatory limits (5 µg/kg and 10 µg/kg, respectively). For all the toxins, a higher occurrence was recorded in 2021 as compared to 2019. In 2019, AFG1 showed the highest content (median value = 27.9 µg/kg), while, in 2021, it was AFB1 (7.1 µg/kg). 

In Malawi, AFB1 was the most prevalent at harvest during the two years of sampling (20.7% in 2019 and 24.3% in 2021), and for all the studied toxins, there was a slightly higher occurrence in 2021 as compared to 2019. The occurrence almost doubled after 6 months of storage, reaching 41.7% for AFG1. Approximately 40% to 70% of the contaminated samples had AFB1 and total aflatoxins above the regulatory limits. For all the sampling rounds in Malawi, AFG1 was the highest, with median values of 5.2, 5.3, and 7.6 µg/kg recorded at harvest in 2019, six months after harvest in 2019, and at harvest in 2021, respectively. 

Association tests (Mann–Whitney test and Spearman’s test at the 0.05 significance level) showed that the education level of the household head, their aflatoxin knowledge, and the microclimatic (agro-ecological) zone where the household is located were the most common factors associated with aflatoxins contamination. In Tanzania, in 2019, higher toxin levels were found in maize from the valley zone, while the dry lowland and higher mountain zones showed no toxin contamination; in 2021, however, very low toxins were found in the valley zone, while the dry lowland and higher mountain zones recorded the highest contamination levels. Similar variations in the aflatoxin contamination levels with microclimatic zones and with the sampling year were also observed in Malawi. These results suggest that a combination of factors should be considered when predicting the aflatoxin contamination of maize.

**Keywords:** aflatoxin B1; maize harvest; storage; microclimatic zone; Sub-Saharan Africa

### 3.5. Mycotoxin Monitoring—South African Maize and Wheat Crop Surveys

LouwWiana[Aff af33-toxins-15-00174]MeyerHannalienThe Southern African Grain Laboratory NPC (SAGL), 477 Witherite Street, Pretoria 0041, South Africa*Correspondence: wiana.louw@sagl.co.za

The Southern African Grain Laboratory is the reference laboratory for the grain industry in South Africa. The lab conducts annual crop quality surveys on commercially produced wheat and maize from different production areas. Mycotoxin monitoring forms an integral part of these surveys conducted in collaboration with the Agricultural Commodity Trusts and the Grain Silo Industry in South Africa. Representative samples of wheat and maize are collected at intake and submitted to the laboratory during each harvest season. Multi-mycotoxin analyses, including thirteen of the most important mycotoxins, are performed on a proportionally selected number of the crop survey samples using an ISO 17025 accredited UPLC-MS/MS multi-mycotoxin method. The mycotoxins monitored are aflatoxin B_1_, B_2_, G_1_, and G_2_; fumonisin B_1_, B_2_, and B_3_ (FUM); deoxynivalenol (DON); 15-acetyl-deoxynivalenol (15-ADON); T-2 toxin; HT-2 toxin; zearalenone (ZON); and ochratoxin A (OTA). The objectives to evaluate the occurrence of mycotoxins in South African wheat and maize and build a reliable database for targeted research and management of the mycotoxin levels have been achieved.

The results presented summarized the year-on-year trends of mycotoxin occurrence over the last eight seasons. Deoxynivalenol and fumonisin were the most predominant mycotoxins in maize, with differences in occurrence and concentrations observed between seasons and production regions. Deoxynivalenol was the only mycotoxin found in wheat, and although the levels were mostly below the regulated levels, an increasing trend was observed in the last two seasons. The details of a post-storage, preprocessing mycotoxin monitoring survey on maize was also presented. It was concluded that mycotoxin survey results over eleven seasons provided a useful South African perspective for commercially produced wheat and maize. The importance for the grain industry to continue monitoring the mycotoxin levels at intake at the processing stage and in the final food and feed products was emphasized.

**Keywords:** mycotoxins; maize; wheat; South Africa; survey

### 3.6. Data Availability and Ideas for Several African Countries Case Study

FaladeTitilayo[Aff af35-toxins-15-00174]Ortega-BeltranAlejandroInternational Institute of Tropical Agriculture, West Africa Hub, PMB 5320, Oyo Road, Ibadan 200001, Oyo State, Nigeria*Correspondence: T.Falade@cgiar.org

An important product from the research is the generation of data that can serve a purpose beyond that which was initially anticipated. In collaboration with national and international partners, the International Institute of Tropical Agriculture has produced research on aflatoxins in many African countries. These georeferenced data (published and unpublished) include aflatoxin incidences in two major staple crops, maize and groundnuts, but also other crops such as sorghum, chili peppers, and sesame. This includes aflatoxin concentrations in samples collected at harvest or postharvest from farmers’ fields, stores, or markets across multiple seasons. In addition, there was fungal diversity data from grains and soil samples from farmers’ fields. The data represent a valuable resource on location- and time-specific aflatoxin incidences and dominant fungi associated with aflatoxin incidences, which can be employed to develop aflatoxin risk modeling and the development of intervention strategies. Members of *Aspergillus* section *flavi* are toxigenic and nontoxigenic. Over 100,000 isolates have been obtained from grains and soils from across 20+ African countries and their toxin production capabilities assessed. Several nontoxigenic isolates have been identified as useful for aflatoxin mitigation interventions. Various bioprotectants under the generic name Aflasafe^®^ have been developed, tested, registered, and transferred to the private sector for use at scale. Reduced (80 to 100% less) aflatoxin contamination results when treating crops with Aflasafe products compared with untreated crops. Georeferenced aflatoxin data, morphological, molecular, and aflatoxin production abilities of isolates recovered from multiple studies in multiple countries and multiple seasons can be utilized for predictive modeling and risk interventions in combination with metadata on agroecological, demographic, and other relevant data for informing risk management strategies.

**Keywords:** aflatoxin; risk modeling; biological control; *Aspergillus*

### 3.7. Limiting Mycotoxin Exposure of Livestock by Monitoring and Forecasting of Contaminations in Feed Crops

SchweigerWolfgang[Aff af37-toxins-15-00174]PlatzerAlexanderJenkinsTimothySchatzmayrGerdBIOMIN Research Center, Technopark 1, 3430 Tulln, Austria*Correspondence: Wolfgang.Schweiger@dsm.com

Mycotoxins negatively affect animal health and, consequently, animal performance. Minimizing mycotoxins in feed improves the animal’s development but also reduces crop waste or the need for blending below harmful levels. Their occurrence in crops may vary strongly depending on the growing region, agronomic factors, and climatic conditions. Measures aiming to reduce contamination in the feed need to be based on the prior knowledge on toxin prevalence and contamination levels in a certain area. The BIOMIN mycotoxin survey is an extensive database that aggregates global mycotoxin occurrence data from over 130,000 samples collected in 75 countries. The survey provides users with a sound basis for estimating their local mycotoxin risk and is now being expanded to include the prediction of mycotoxins in upcoming harvests by using weather as the main predictor. Users can either access field-based custom predictions for deoxynivalenol on wheat via the MyToolBox platform (https://mytoolbox.dsm.com/; accessed on 7 September 2022) or access weekly updated predictions for maize and wheat in combination with four regulated mycotoxins for regions in the country of their choice. An increased regional risk may warrant targeted toxin-specific measures, ranging from changed agronomic practices to additional testing of grains at risk or the treatment of finished feed with mycotoxin-degrading products, such as Biomin’s Mycofix product line.

**Keywords:** mycotoxins; deoxynivalenol; wheat; risk modeling; animal health; MyToolBox

## Data Availability

Not applicable.

