# Peer review of "Abstracts of Presentations to the Working Session on Improving Predictive Modeling of Mycotoxin Risk for Africa Held at the 3rd ASM2022 on 7 September 2022, in Stellenbosch, South Africa"

_toxins, 2023, doi:10.3390/toxins15030174_

Round 1

Reviewer 1 Report

I have reviewed the conference report entitled “Abstracts of contributions to the work session on improving predictive modelling of mycotoxin risk for Africa held at the 3rd ASM2022 on September 7, 2022 in Stellenbosch, South Africa”.

After my review of the report, it is my judgment that it is not a well-designed work, and there are many flaws in this report, such as grammars, units, and font. I suggest the authors to reorganize the manuscript carefully if they would like to publish this report. The manuscript needs careful editing and particular attention to English grammar and sentence structure.

According to the lectures by many experts, it seems that the APHLIS model and other prediction model could predict risk of aflatoxin contamination but only to some extent. Frankly speaking, in my opinion, I do not think the prediction models that built based on one or two factors could be used for accurate risk warning of mycotoxin contamination in nature. We know that mycotoxin biosynthesis is affected by many factors, such as sunlight, temperature, humidity, and rainfall, etc., and a combination of these factors should be considered. According to the lectures, to achieve these, we still have a long way to go.

The current version of the report is just a draft, and there are many things to be mentioned or clarified in the report. All the marked places in the PDF version must be revised.

There are some examples of my specific comments as follow.

1.     I strongly suggest the authors to rewrite the “Abstract”, the current version is completely copied from the main text. As an officially published report, it is not stringent enough.

2.     What is the differences of “APHILIS” and “APHLIS”?

3.     Please explain all the abbreviations, such as ASAP in Line 118, AFLA-maize and FER-maize in Line 224, and SA in Ling 319, etc.

4.     What is the differences of “AG1” and “AFG1”, are they different mycotoxins?

5.     Line 193, the three zones should be provided.

6.     Lines 195-199, I do not think “ppb” is suitable for publication.

7.     Line 265, the expression is inaccurate.

8.     The research paper and oral presentation are different. Some sentences should be rewrite or reorganized in the report. Such as Lines 79-81, Lines 333-337, I suggest the authors to rewrite these sentences.

9.     I suggest the authors to provide an introduction of the conference in addition to the oral presentations. Besides, what’s the significance of the conference? It’s better to emphasize the main contribution of this conference in the report.

Author Response

I have reviewed the conference report entitled “Abstracts of contributions to the work session on improving predictive modelling of mycotoxin risk for Africa held at the 3rd ASM2022 on September 7, 2022 in Stellenbosch, South Africa”.

After my review of the report, it is my judgment that it is not a well-designed work, and there are many flaws in this report, such as grammars, units, and font. I suggest the authors to reorganize the manuscript carefully if they would like to publish this report. The manuscript needs careful editing and particular attention to English grammar and sentence structure.

Answer: The text has been completely reorganized and revised by 2 native English speakers

According to the lectures by many experts, it seems that the APHLIS model and other prediction model could predict risk of aflatoxin contamination but only to some extent. Frankly speaking, in my opinion, I do not think the prediction models that built based on one or two factors could be used for accurate risk warning of mycotoxin contamination in nature. We know that mycotoxin biosynthesis is affected by many factors, such as sunlight, temperature, humidity, and rainfall, etc., and a combination of these factors should be considered. According to the lectures, to achieve these, we still have a long way to go.

We agree with the Review, this was one of the reasons behind organizing the workshop and the result was that we need to move forward precisely to fill the current gaps

The current version of the report is just a draft, and there are many things to be mentioned or clarified in the report. All the marked places in the PDF version must be revised.

Answer: The text has been completely reorganized and revised by 2 native English speakers

There are some examples of my specific comments as follow.

  1. I strongly suggest the authors to rewrite the “Abstract”, the current version is completely copied from the main text. As an officially published report, it is not stringent enough. The Abstract has been rewritten
  2. What is the differences of “APHILIS” and “APHLIS”? Typing error
  3. Please explain all the abbreviations, such as ASAP in Line 118, AFLA-maize and FER-maize in Line 224, and SA in Ling 319, etc. Done: ASAP (Anomaly Hotspots of Agricultural Production). At European level, two weather-based mechanistic models were developed, ‘AFLA-maize’ and ‘FER-maize’, predicting aflatoxin B1 and fumonisins, respectively.
  4. What is the differences of “AG1” and “AFG1”, are they different mycotoxins? Aflatoxins B1, B2, G1, and G2
  5. Line 193, the three zones should be provided. Done. Savanna (in Brong Ahafo and Northern Regions), Humid Forest (in Ashanti and part of Brong Ahafo Regions), and Southern Guinea Savanna (in Upper East and Upper West Regions.
  6. Lines 195-199, I do not think “ppb” is suitable for publication.

Answer: Parts per billion (ppb) is the number of units of mass of a contaminant per 1000 million units of total mass. Also µg/L or micrograms per liter.

  1. Line 265, the expression is inaccurate. Rewritten
  2. The research paper and oral presentation are different. Some sentences should be rewrite or reorganized in the report. Such as Lines 79-81, Lines 333-337, I suggest the authors to rewrite these sentences. Rewritten
  3. I suggest the authors to provide an introduction of the conference in addition to the oral presentations. Besides, what’s the significance of the conference? It’s better to emphasize the main contribution of this conference in the report. Rewritten

Reviewer 2 Report

The manuscript provides a report on the conference session on enhancing predictive modelling of mycotoxin risk for Africa held at the 3rd ASM2022 in Stellenbosch, South Africa on September 7, 2022. The report provides a preface and an introduction to the session as well as captures the abstracts presented during this session of the conference. Overall, the report is well written. Specific comments on the report can be found below.

1.      The acknowledgment should be moved to the end of the report.

2.      A concluding summary that synthesises all the information and places it in context, focusing on the two objectives of the conference session indicated in the abstract and introduction should be provided.

Author Response

The manuscript provides a report on the conference session on enhancing predictive modelling of mycotoxin risk for Africa held at the 3rd ASM2022 in Stellenbosch, South Africa on September 7, 2022. The report provides a preface and an introduction to the session as well as captures the abstracts presented during this session of the conference. Overall, the report is well written. Specific comments on the report can be found below.

  1. The acknowledgment should be moved to the end of the report. Answer: Done

  1. A concluding summary that synthesises all the information and places it in context, focusing on the two objectives of the conference session indicated in the abstract and introduction should be provided.

Answer: The text has been rewritten and structured according to the Editor's instructions

Reviewer 3 Report

Dear authors,

The data presented in this paper regarding food safety in mycotoxins are of interest even if the approach is limited, its feasibility is still difficult to assess and requires more testing.

The results presented may be published.

Author Response

The data presented in this paper regarding food safety in mycotoxins are of interest even if the approach is limited, its feasibility is still difficult to assess and requires more testing.

The results presented may be published.

Answer: Thank you for giving us this chance

Round 2

Reviewer 1 Report

I appreciate the authors for their responses to my comments. The authors have improved the manuscript on writing, but there are still some obvious mistakes in the main text and my specific comments as follow.

Specific comments:

1.      Line 6, change “warnings” to “warning”.

2.      Line 43, change “2. Preface” to “1. Preface”, also the numbers of other sections should be revised.

3.      Line 56, 67, and 104, change “aflatoxins” to “aflatoxin”.

4.      Line 57, instead of “To do this” with “To achieve this”.

5.      Change “mycotoxins” to “mycotoxin” in Lines 60, 64, 70, 181, and 184.

6.      Lines 11 and 87, I suggest the authors to double check how many sub-Sahara countries were included in the APHLIS project, 37 or 38?

7.      Line 87, change “oats” to “oat”.

8.      Line 118, change “in storages” to “during storages”.

9.      Line 139, change “about role of” to “about the roles of”.

10.  Line 147, change “(SPI >1.5 SD anomaly” to “(SPI >1.5 SD anomaly)”.

11.  Line 153, delete “the”.

12.  About the mycotoxin concentration units used in the report, ppb (Lines 246 to 251) and μg/kg (Lines 334 to 343), I suggest the authors to choose one of them and μg/kg was recommended.

13.  Moreover, the same abbreviations of different mycotoxins, such as aflatoxins B1, B2, G1, and G2 in Line 330 and aflatoxin B1, B2, G1 and G2 in Line 372 should be used, choose one of them.

Author Response

I appreciate the authors for their responses to my comments.

Answer: Thanks

 The authors have improved the manuscript on writing, but there are still some obvious mistakes in the main text and my specific comments as follow.

Answer: All comments have been accepted

Specific comments:

  1. Line 6, change “warnings” to “warning”. Done
  2. Line 43, change “2. Preface” to “1. Preface”, also the numbers of other sections should be revised. Done
  3. Line 56, 67, and 104, change “aflatoxins” to “aflatoxin”. Done
  4. Line 57, instead of “To do this” with “To achieve this”. Done
  5. Change “mycotoxins” to “mycotoxin” in Lines 60, 64, 70, 181, and 184. Done
  6. Lines 11 and 87, I suggest the authors to double check how many sub-Sahara countries were included in the APHLIS project, 37 or 38? Done 37
  7. Line 87, change “oats” to “oat”. Done
  8. Line 118, change “in storages” to “during storages”. Done
  9. Line 139, change “about role of” to “about the roles of”. Done
  10. Line 147, change “(SPI >1.5 SD anomaly” to “(SPI >1.5 SD anomaly)”. Done
  11. Line 153, delete “the”. Done
  12. About the mycotoxin concentration units used in the report, ppb (Lines 246 to 251) and μg/kg (Lines 334 to 343), I suggest the authors to choose one of them and μg/kg was recommended. Done
  13. Moreover, the same abbreviations of different mycotoxins, such as aflatoxins B1, B2, G1, and G2 in Line 330 and aflatoxin B1, B2, G1 and G2 in Line 372 should be used, choose one of them. Done